# Important features of retail shoes for women with rheumatoid arthritis: A Delphi consensus survey

Peta Ellen Tehan[1,2]*, William J. Taylor[3], Matthew Carroll[2], Nicola Dalbeth[4,5], Keith Rome[2]

**1** School of Health Sciences, Faculty of Health and Medicine, University of Newcastle, Newcastle, NSW, Australia, **2** School of Clinical Sciences, Faculty of Health and Environmental Sciences, Auckland University of Technology, Auckland, New Zealand, **3** Department of Medicine, University of Otago, Wellington, New Zealand, **4** School of Medicine, Faculty of Medical and Health Sciences, University of Auckland, Auckland, New Zealand, **5** Department of Rheumatology, Auckland District Health Board, Auckland, New Zealand

* Peta.Tehan@newcastle.edu.au

## Abstract

### Objectives

Footwear management aims to preserve foot function, reduce the burden of foot pain and maintain joint mobility in women with rheumatoid arthritis (RA). Whilst retail footwear is commonly recommended by health professionals, there is no current consensus on recommended features of retail footwear for women with RA. This study aimed to determine consensus from health professionals about the important features of retail footwear for women with RA.

### Methods

An international Delphi exercise using online survey software was conducted with 39 participants from health care backgrounds. Three iterative rounds were conducted. In the first round, participants listed features of retail footwear that would be important for women with RA. Responses of the first round, combined with results of a scoping review of patient-reported outcome measures used in assessing footwear in arthritis and a qualitative analysis of female patients' perspectives of retail footwear in RA were used to create items for the second round. Items were scored by a 9-point rating scale with consensus defined by the RAND/UCLA disagreement index. The third round consisted of items which did not reach consensus or scored >1 on the RAND/UCLA disagreement index from round two.

### Results

Fifty-eight items (n = 58) were generated for rating and at the end of three iterative rounds, there was agreement that thirty-eight items were important, that two were not important, and there was no agreement for a further eighteen items. Item themes reaching consensus included footwear characteristics and acceptability and psychosocial aspects of footwear. Footwear characteristics related to heel height, shape, cushioning, toe box size, adjustable

**Data Availability Statement:** All relevant data are within the manuscript and its Supporting Information files.

**Funding:** The authors received no specific funding for this work.

**Competing interests:** PT, MC, WT have declared no competing interests exist. KR has received funding from ASICS, outside the submitted work. Prof Dalbeth reports research grant funding from Amgen and AstraZeneca, and speaker fees from Pfizer Inc, Horizon, Janssen Pharmaceuticals, and AbbVie, as well as consulting fees from Horizon, Hengrui, and Kowa, outside the submitted work. This does not alter our adherence to PLOS ONE policies on sharing data and materials.

fastening, removable insoles, mid-foot support and soft accommodative uppers. Acceptability and psychosocial aspects included affordability, comfort, aesthetic, style, colour and impact on femininity.

## Conclusion

This consensus exercise has identified the important features of retail footwear for women with RA.

## Background

Rheumatoid arthritis (RA) is a common form of inflammatory arthritis affecting women, with the feet often affected [1, 2]. Non-pharmacological management of foot pain in women with RA is centered around preservation of foot function, reducing pain and maintaining mobility [3]. Footwear is the most common non-pharmacological treatment modality used to achieve these goals, with a correctly fitted and functional shoe being therapeutically valuable [3–5]. Conversely, a poorly fitted or inappropriate shoe can exacerbate foot pain, reduce mobility and impact overall quality of life [6]. Therapeutic (medical grade or custom made) footwear is often prescribed for women with RA, however is frequently unavailable to many women with RA due to cost [6]. Furthermore, therapeutic footwear is often not accepted by women, which reduces wearing adherence and therefore the opportunity to achieve therapeutic benefit is lost [7]. Lack of adherence with therapeutic footwear is multi-factorial, often compounded by a lack of acceptance of their underlying condition, and a disparity in expectations of the footwear [3, 7, 8]. This non-acceptance of therapeutic footwear results in many women seeking retail footwear as an alternative.

Women with RA have particular footwear requirements, with many presenting with foot pain and deformity which makes fitting retail footwear difficult [4]. Furthermore, women associate aspects of their identity and femininity with their footwear choices [8, 9]. The presence of RA can negatively impact on choice, and an ideal shoe should accommodate deformity and offer comfort, whilst offering the women choices in style [8]. Many women will look to health professionals for advice when seeking appropriate footwear [8, 10], however currently there is no consensus on what features of retail footwear are important. This Delphi survey aimed to reach consensus from health professionals about what specific retail footwear features are important for women with RA.

## Method

### Expert identification

A sample of health professionals were identified by the researchers (PT and KR) from the United States of America, Europe, Australia and New Zealand, with expertise in RA and footwear. Participants were identified through professional networks, including New Zealand Rheumatology Association, Podiatric Rheumatology Care Association (UK), literature and guideline searching. Seventy-five (n = 75) were approached in writing and invited to participate. This number allowed for drop out at each stage. Dando [10] reported that the inclusion of a wide range of professions from a range of clinical backgrounds with a geographical diversity is good practice as it develops the participants to be a representative group [11]. A heterogenous group, with a range of stakeholders, encourages different outlooks and decision-making, which in turn enriches the data leading to better outcomes of credibility and

acceptability [12]. The panel inclusion criteria were health practitioners or researchers with specialised knowledge in rheumatology and/or footwear prescription. This study received ethical approval from the Health and Disability Ethics Committee, New Zealand (17/NTB/3/AM03). All panel members provided informed electronic consent prior to participation.

## Data collection

The Delphi technique was used to elicit the opinions of participating health professionals through three successive rounds of iterative questionnaires using online survey software (Qualtrics XM, Provo, UT) to identify specific retail footwear features that were important for women with RA. Responses to each round of the Delphi were collated, analysed and the results returned to participants in the form of another questionnaire until an acceptable degree of consensus was achieved [13, 14]. Reminders to participants were sent via email one week following the opening of the survey round, and participants were given an additional week to complete the survey before being classified as a non-responder.

The first round consisted of demographic and descriptive questions followed by one open-ended question, asking participants to list up to ten specific footwear features that they advised women with RA to look for when seeking retail footwear. Items for round two were generated based on the open-ended results of round one combined with additional items from the results of a scoping review which mapped the literature assessing footwear in arthritis [15], and a qualitative analysis of female patients with RAs' perspectives on retail footwear [8]. In round two, participants were asked to rate fifty-eight (n = 58) items relating to retail footwear features for women with RA (S1 Table). All participants used a 9-point rating scale (1 = not at all important; 9 = absolutely essential) [16]. Consensus was defined by the RAND/University of California Los Angeles disagreement index (RAND/UCLA DI) whereby values >1 indicated panel disagreement or extreme variation in findings [16]. The lower the RAND/UCLA DI, the lower the level of disagreement (i.e. the higher the level of agreement) [16]. Medians, 30th and 70th percentile ranges were calculated for each item. Items which had median scores of between four and six, or where there was disagreement (RAND/UCLA DI > 1) were considered lacking consensus and were taken back to participants for further consideration in round three. Items where the median score was between seven and nine and RAND/UCLA DI <1 were considered agreed important, and items with a median score between one and three and a RAND/UCLA DI <1 were considered agreed not important. In round three, each item also included the group median rating and 30-70th percentile range results from round two. All participants were emailed a copy of their individual responses from round two. The third round gave participants opportunity to change their answers in light of the group's average. If the median score on items without consensus remained unchanged between rounds, the items were confirmed as lacking consensus.

## Data analysis

Demographic data for all participants was collected and analysed using IBM Statistical Packages for the Social Sciences (SPSS) Version 25 for Windows (SPSS Inc., Armonk, NY, USA). Means and standard deviations were calculated for descriptive data, and for content items, medians and 30th - 70th percentile ranges were calculated.

## Results

### Delphi panel

Thirty-nine (n = 39) participants responded to the first round (Table 1), seventeen (n = 44%) were female with mean (SD) of 23 (10) years since qualifying as a health professional. The

**Table 1. Descriptive data of Delphi panel.**

| | | Round 1 | Round 2 | Round 3 |
|---|---|---|---|---|
| Invites (n) | | 75 | 39 | 33 |
| Responses (n) | | 39 | 33 | 30 |
| Completion rate (%) | | 44 | 85 | 91 |
| Sex n (%) | Female | 22 (56) | 20 (61) | 19 (58) |
| | male | 17 (44) | 13 (39) | 11 (37) |
| Podiatrist n (%) | | 21 (54) | 17 (52) | 16 (53) |
| Rheumatologist n (%) | | 12 (31) | 11 (33) | 10 (33) |
| Physiotherapist n (%) | | 1 (3) | 1 (3) | 1 (3) |
| Nurse n (%) | | 2 (5) | 2 (6) | 1 (3) |
| Rehabilitation medicine researcher n (%) | | 1 (3) | 0 | 0 |
| Rheumatology researcher n (%) | | 2 (5) | 2 (6) | 2 (7) |
| Years in practice mean years, (SD) | | 23.10 (9) | 24.03 (9) | 22.73 (10) |
| Location (n, %) | Europe | 19 (49) | 15 (46) | 13 (43) |
| | USA | 3 (8) | 3 (9) | 3 (10) |
| | Australia | 6 (15) | 6 (18) | 6 (20) |
| | New Zealand | 11 (28) | 9 (27) | 8 (27) |

majority of participants were podiatrists (n = 21, 54%), but also included rheumatologists (n = 12, 31%), a physiotherapist (n = 1, 3%), rheumatology nurses (N = 2, 5%), a rehabilitation physician (n = 1, 3%) and health-related researchers (n = 2, 5%). The majority of participants were based in the United Kingdom (n = 16, 41%). Thirty-three (n = 33) responses were received for the second round, and thirty (n = 30) in the third round.

## Results of Delphi survey

Fifty-one items (n = 51) were generated from the panel following round one. Themes were developed and included: general footwear fit and acceptability, specific footwear characteristics including heel, rearfoot, midfoot, forefoot, upper, outsole, fixation, and miscellaneous. These items, were taken to the panel in round two (n = 58). There was agreement that two items were not important; shoes should be one size larger than required (median 2 IPR 2–4); and shoes should be slip-on (median 3.0 IPR 2–5). Thirty-six (n = 36) items were agreed as important and twenty (n = 20) did not achieve consensus (Table 2). One item from round two (women should own two pairs of shoes which can be alternated) had a RAND/UCLA DI of 1.62 indicating disagreement and was taken back to the panel in round three for further consideration. Following round three, a further two items were accepted and eighteen (n = 18) did not reach consensus with minimal change between median scores from round two to round three, with all items RAND/UCLA DI index scores under one indicating no disagreement. Therefore, the Delphi survey was closed.

Six items had a median score of 9 indicating strong agreement from the panel as absolutely essential important features of retail footwear. These included shoes should be affordable (median 9 IPR 8–9), shoes should be deemed comfortable by the owner (median 9 IPR 9–9), shoes should be deemed satisfactory by the owner (median 9 IPR 8–9), shoes should not have seams over pressure areas of deformities (median 9 IPR 7.2–9), shoes should avoid pressure points in the foot (median 9 IPR 8–9) and shoes should be appropriately sized, with adequate length and width (median 9 IPR 9–9). All items reaching agreement are presented in Table 3.

The content analysis of items generated by the panel revealed three broad themes, footwear characteristics, which included items relating to the construction of the shoe and its

anatomical features. Items such as: shoes should have a soft accommodative upper, shoes should have a sturdy rearfoot and shoes should have a heel height of no greater than 3cm were contained within this theme. The second theme, footwear acceptability related to women's personal preferences and satisfaction. Items such as shoes should be available in a range of colours and styles, and shoes should be affordable were contained within this theme. The third theme, the psychosocial aspects of footwear, contained items such as shoes should look feminine and shoes should not cause self-consciousness.

Important items included those relating to sizing and fit, aesthetic, satisfaction, comfort, cushioning, fastening, heel height, toe box shape, material, weight and insole removability (Table 2). The panel agreed that shoes should be appropriately sized, with adequate length and width, be deep enough to accommodate an orthoses/insert, not have seams over pressure areas or deformities, and avoid pressure points in the foot.

There was consensus that shoes should be aesthetically pleasing, be available in a range of colours and styles, be deemed acceptable by the owner, look feminine, not cause self-consciousness, be appropriate for the social occasion and activity appropriate. There was also agreement that comfort, cushioning (in the heel, midfoot and forefoot) and affordability were very important. Important footwear characteristics relating to the rearfoot of the shoe included: a sturdy rearfoot, cushioned heel, broad heel base, heel height of no greater than 3 cm and a heel counter which conforms to the geometry of the calcaneus. Adequate midfoot support, metatarsal support and longitudinal arch support were all accepted as important mid-foot features of a retail shoe. Important forefoot features included a toe box which was wide and deep, and able to accommodate toes. Other features included fastenings which were adjustable, well secured, and easy to put on and take off. Footwear made of breathable material, a conformable upper, textured outsole, lightweight with adequate shock absorption were also accepted as important features.

There was no agreement about the importance of the following items after round three: need for professional fitting, needing to own two pairs of shoes which can be alternated; that comfort should be preferable to aesthetic. Agreement was not reached about the importance of specific rearfoot footwear features including the need for a shoe to contain a heel stability cup, a stiff heel counter and absence of a sling back. Agreement was not reached about the importance of midfoot and forefoot features including a metatarsal rocker, adjustability in metatarsal areas, a round toe box and a semi-rigid shank; and other general features including leather upper, supportive inlay, Velcro® fastenings, lace-up fastening, containing features found in running shoes, made from soft fabric and a boot style if support is required.

## Discussion

This Delphi exercise sought to generate agreement between health care professionals about important features of retail footwear for women with RA. The panel of expert clinicians and researchers reached consensus on thirty-eight items which were considered important (Table 3). The footwear characteristics identified by the panel had two overarching themes. Footwear characteristics, which included items related to the construction of the shoe and its anatomical features. The second theme, acceptability and the psychosocial aspects of footwear, included items relating to women's personal preferences and satisfaction.

Consistent with previous research of patient perspectives on retail footwear, items such as comfort, aesthetic, ease of fastening, weight, and upper material reached consensus as important [8, 9, 17, 18]. However, unique to this study, were items relating to shoe design such as the need for mid-foot support, metatarsal support and longitudinal arch support. Although health professionals may be able to assess footwear for these features, many women with RA would

**Table 2. Results of the Delphi exercise for important features of retail footwear for women with RA.**

| Footwear Item Category | Item | Median score (30–70 IPR)[A] | DI[A] |
|---|---|---|---|
| **General Shoe Fit** | *Agreed Important (round*)* | | |
| | Shoes should be appropriately sized, with adequate length and width (2) | 9 (9–9) | 0 |
| | Shoes should be deep enough to accommodate an orthoses/insert (2) | 8 (7–8.8) | 0.26 |
| | Shoes should not have seams over pressure areas or deformities (2) | 9 (7.2–9) | 0.21 |
| | Shoes should avoid pressure points in the foot (2) | 9 (8–9) | 0.11 |
| | Shoe shape should be relevant to foot shape (2) | 8 (7–9) | 0.29 |
| | *Agreed Not Important (round*)* | | |
| | Shoes should be one size larger than required (2) | 2.0 (2–4) | 0.29 |
| | *Did not reach consensus* | | |
| | Women with RA should be professionally fitted for their footwear | 6.0 (6–7) | 0.25 |
| | Women should own two pairs of shoes which can be alternated | 5.5 (5–7) | 0.64 |
| **Aesthetic** | *Agreed Important (round*)* | | |
| | Shoes should be aesthetically pleasing (2) | 7 (7–8) | 0.18 |
| | There should be a range of choice in styles available (2) | 8 (7–8.8) | 0.26 |
| | There should be a range of choice in colours available (2) | 8 (7–8) | 0.14 |
| | Shoes should be deemed satisfactory by the owner (2) | 9 (8–9) | 0.11 |
| | Shoes should look feminine (2) | 7 (6–7) | 0.18 |
| | Shoes should not cause self-consciousness (2) | 8 (8–9) | 0.14 |
| | Shoes should be appropriate for the social occasion (2) | 7 (6.2–8) | 0.33 |
| | *Did not reach consensus* | | |
| | Shoe comfort should be preferable to aesthetic | 6.0 (5–7) | 0.25 |
| **Specific Features -Rearfoot/Heel** | *Agreed Important (round*)* | | |
| | Shoes should have a sturdy rearfoot (2) | 7 (6–8) | 0.37 |
| | Shoes should have a heel height of no greater than 3 cm (1.5 inches) (2) | 7 (6–8) | 0.37 |
| | Shoes should have a cushioned heel (2) | 7 (6–7) | 0.18 |
| | Shoes should have a broad heel base (2) | 7 (6–8) | 0.37 |
| | Shoes should contain a heel counter which conforms to the geometry of the calcaneus (2) | 7 (5–8) | 0.56 |
| | *Did not reach consensus* | | |
| | Shoes should not contain a sling back | 6.0 (5–7) | 0.25 |
| | Shoes should contain a heel stability cup | 6.0 (6–7) | 0.25 |
| | Shoes should have a stiff heel counter | 6.0 (6–7) | 0.25 |
| **Specific Features -midfoot/forefoot** | *Agreed Important (round*)* | | |
| | Shoes should contain adequate midfoot support (2) | 7 (6.2–8) | 0.39 |
| | Shoes should have a wide toe box (2) | 8 (7–8) | 0.89 |
| | Shoes should have a deep toe box (2) | 7 (6–8) | 0.37 |
| | Shoes should contain forefoot to rearfoot cushioning (2) | 7 (6–7.1) | 0.5 |
| | Shoes should be able to accommodate the toes (2) | 8 (8–9) | 0.77 |
| | Shoes should have longitudinal arch support (3) | 7.0 (6–7) | 0.18 |
| | Shoes should contain metatarsal support (3) | 7.0 (6–7) | 0.18 |
| | *Did not reach Consensus* | | |
| | Shoes should contain a semi-rigid shank | 6.0 (6–6) | 0 |
| | Shoes should have a rocker in metatarsal areas | 6.0 (5–6) | 0.25 |
| | Shoes should be adjustable in metatarsal areas | 6.0 (6–7) | 0.25 |
| | Shoes should have a round toe box | 6.0 (6–7) | 0.25 |

(*Continued*)

**Table 2.** (Continued)

| Footwear Item Category | Item | Median score (30–70 IPR)$^A$ | DI$^A$ |
|---|---|---|---|
| **Specific Features -Outsole/Upper/Insole** | *Agreed Important (round*\*)* | | |
| | Shoes should have an outsole which is textured and non-slip (2) | 8 (7–8) | 0.89 |
| | Shoes should have a soft accommodative upper (2) | 7 (6–8) | 0.37 |
| | Shoes should have an upper which is conformable (2) | 8 (8–9) | 0.77 |
| | Shoes should contain an insole which is removable (2) | 8 (6–9) | 0.43 |
| | *Did not reach Consensus* | | |
| | Shoes should have a thick outsole | 6.0 (5–6) | 0.25 |
| | Shoes should have a leather upper | 5.0 (4–5) | 0.42 |
| | Shoes should contain a supportive inlay | 6.0 (6–7) | 0.25 |
| **Specific Features-Fixation** | *Agreed Important (round*\*)* | | |
| | Shoes should be well secured (2) | 8 (7–9) | 0.37 |
| | Shoes should be easy to put on and take off (2) | 8 (8–9) | 0.77 |
| | Shoes should have fastenings which are adjustable (2) | 8 (6–9) | 0.43 |
| | *Agreed Not Important (round*\*)* | | |
| | Shoes should be slip on (2) | 3.0 (2–5) | 0.29 |
| | *Did not reach Consensus* | | |
| | Shoes should have Velcro fastenings | 5.0 (5–5) | 0 |
| | Shoes should be lace up | 5.0 (4–5) | 0.42 |
| | Shoes should have Velcro fastenings | 5.0 (5–5) | 0 |
| **Specific Features -Other** | *Agreed Important (round*\*)* | | |
| | Shoes should provide adequate shock absorption (2) | 8 (6–8) | 0.56 |
| | Shoes should be activity appropriate (2) | 8 (8–9) | 0.77 |
| | Shoes should be made of breathable material (2) | 7 (5.2–8) | 0.30 |
| | Shoes should be suitable for the weather conditions (warm in winter, cool in summer) (2) | 8 (7–9) | 0.37 |
| | Shoes should be lightweight (2) | 7 (6–8) | 0.37 |
| | Shoes should be deemed comfortable by the owner (2) | 9 (9–9) | 0 |
| | Shoes should be affordable (2) | 9 (8–9) | 0.11 |
| | *Did not reach consensus* | | |
| | Shoes should be made from soft fabric | 6.0 (5–6) | 0.25 |
| | Shoes should be a boot style if additional support is required | 6.0 (5–6) | 0.25 |
| | Shoes should contain features found in trainers/running shoes | 6.0 (6–7) | 0.25 |

$^A$A RAND/UCLA disagreement index of > 1 indicates panel disagreement, a RAND/UCLA DI value of < 1 indicates panel agreement

\*- round item reached consensus

Median scores are representative of items scored on a 9 point scale, with 1 indicating an item was 'not at all important', and 9 indicating an item was 'absolutely essential'.

Items were accepted with a median score between 7 and 9 and a RAND/UCLA DI value of <1, Items were rejected with a median score between 1 and 3 and a RAND/UCLA DI value of <1. Items were considered lacking consensus with a mean value between 4 and 6 and/or a RAND/UCLA DI value of >1

IPR–interpercentile range

need to receive information about how to identify retail footwear which contains these specific features. Furthermore, many of these features would not be visible, so some understanding of footwear materials and design would be needed to determine if these aspects were present. Footwear manufacturers, retailers and retail staff would also need to make information concerning footwear characteristics readily available and presented in a manner which is easily understood. However, simpler design aspects which were agreed upon such as heel height,

**Table 3. Important features of retail footwear for women with RA—Items reaching agreement.**

| Content Item Theme | Item |
|---|---|
| **Footwear Characteristics** | Shoes should be appropriately sized, with adequate length and width |
| | Shoes should be deep enough to accommodate an orthoses/insert |
| | Shoes should not have seams over pressure areas or deformities |
| | Shoes should avoid pressure points in the foot |
| | Shoe shape should be relevant to foot shape |
| | Shoes should have a sturdy rearfoot |
| | Shoes should have a heel height of no greater than 3 cm (1.5 inches) |
| | Shoes should have a cushioned heel |
| | Shoes should have a broad heel base |
| | Shoes should contain a heel counter which conforms to the geometry of the calcaneus |
| | Shoes should contain adequate midfoot support |
| | Shoes should have a wide toe box |
| | Shoes should have a deep toe box |
| | Shoes should contain forefoot to rearfoot cushioning |
| | Shoes should be able to accommodate the toes |
| | Shoes should have longitudinal arch support |
| | Shoes should contain metatarsal support |
| | Shoes should have an outsole which is textured and non-slip |
| | Shoes should have a soft accommodative upper |
| | Shoes should have an upper which is conformable |
| | Shoes should contain an insole which is removable |
| | Shoes should be well secured |
| | Shoes should be easy to put on and take off |
| | Shoes should have fastenings which are adjustable |
| **Acceptability** | Shoes should provide adequate shock absorption |
| | Shoes should be made of breathable material |
| | Shoes should be suitable for the weather conditions (warm in winter, cool in summer) |
| | Shoes should be lightweight |
| | Shoes should be deemed comfortable by the owner |
| | Shoes should be affordable |
| | There should be a range of choice in styles available |
| | There should be a range of choice in colours available |
| **Psychosocial Aspects** | Shoes should be aesthetically pleasing |
| | Shoes should be activity appropriate |
| | Shoes should be deemed satisfactory by the owner |
| | Shoes should look feminine |
| | Shoes should not cause self-consciousness |
| | Shoes should be appropriate for the social occasion |

fixation method, and outsole material may be more easily communicated to patients due to their visibility. Some of these shoe features have also been previously identified as key footwear design features for different populations, such as older adults [19].

Important features of retail footwear for women with RA extends beyond physical design and technical characteristics. The wearer of the shoe, their opinions and perceptions are important, which has been identified in previous research, and also in the current study. Items relating to acceptability including affordability, comfort and aesthetic were identified as important features by the panel, consistent with previous research investigating RA patient

perceptions of footwear [3, 8, 9]. Furthermore, the psychosocial aspects of footwear including femininity, self-consciousness and appropriateness for the social occasion were also confirmed as important features [3, 8, 9]. This confirms the importance of these features and how patient opinion should be carefully considered and integrated into the footwear management plan. Previous research has confirmed the psychosocial aspects of footwear impact on wear habits, that if women with RA possess footwear which they deem unsatisfactory in any way, they are more likely to restrict their wear time and potentially not gain the potential benefits of use [3, 6].

People with RA often wear footwear lacking supportive mechanisms, cushioning, and protection of toe regions. This may be reflective of the difficulties people with RA face in obtaining retail footwear which suits their needs [18]. Furthermore, when seeking advice on footwear from health professionals, women with RA do not find this advice useful [8]. Comfort is a consistent theme in the literature relating to footwear in RA, with patients consistently seeking comfort from their footwear [8, 9]. This was further confirmed as an important feature by the Delphi panel who agreed that comfort was absolutely essential. Health professionals have a responsibility to support and guide women with RA about what footwear features are important, and what will give them the best chance at achieving comfort.

The results of this study should be considered in light of some limitations. Although every effort was made to recruit a broad range of participants, some experts may have been omitted and their opinions not included. The recommendations of the panel may not be generalizable to different countries due to cultural differences. Furthermore patients were not included in this Delphi study, however, a previous research paper which explored women with RA's experience and preferences of footwear and items from this paper were included in items which were rated by the panel [8].

## Future recommendations

Based on the results of this Delphi survey, future studies should aim to validate the recommendations made by the panel, determine how these findings align with patients' views, and development of a patient reported outcome measure for retail footwear for use by clinicians and researchers should be considered. Furthermore, footwear manufacturers should take note of specific features which were considered important for women with RA and integrate these into future footwear designs.

## Conclusion

The expert panel identified 38 specific features that are important in retail footwear for women with RA. Retail footwear for women with RA should be affordable, comfortable, aesthetically pleasing, and also have characteristics such as a wide toe box, a conformable upper, removable insole, and adjustable fastenings. Health professionals should consider these important features when recommending retail footwear to women with RA.

## Supporting information

**S1 Table. Supporting Information.**
(DOCX)

## Author Contributions

**Conceptualization:** Keith Rome.

**Data curation:** Peta Ellen Tehan.

**Formal analysis:** Peta Ellen Tehan.

**Investigation:** Peta Ellen Tehan.

**Methodology:** Peta Ellen Tehan, William J. Taylor, Nicola Dalbeth, Keith Rome.

**Project administration:** Peta Ellen Tehan, Matthew Carroll, Nicola Dalbeth, Keith Rome.

**Resources:** Keith Rome.

**Software:** Matthew Carroll.

**Supervision:** William J. Taylor, Matthew Carroll, Nicola Dalbeth, Keith Rome.

**Writing – original draft:** Peta Ellen Tehan.

**Writing – review & editing:** Peta Ellen Tehan, William J. Taylor, Matthew Carroll, Nicola Dalbeth, Keith Rome.

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
