## [Decision Letter · Decision Letter 0]

29 Nov 2019

PONE-D-19-30180

Important features of retail shoes for women with rheumatoid arthritis: A Delphi consensus survey

PLOS ONE

Dear Dr. Tehan,

Thank you for submitting your manuscript to PLOS ONE. After careful consideration, we feel that it has merit but does not fully meet PLOS ONE’s publication criteria as it currently stands. Therefore, we invite you to submit a revised version of the manuscript that addresses the points raised during the review process.

The manuscript has been reviewed by two referees.

As you can see from their comments, both reviewers were satisfied with the quality of the study and raised only minor issues that should be easy to improve in a revised version.

We would appreciate receiving your revised manuscript by Jan 13 2020 11:59PM. To enhance the reproducibility of your results, we recommend that if applicable you deposit your laboratory protocols in protocols.io, where a protocol can be assigned its own identifier (DOI) such that it can be cited independently in the future. For instructions see: http://journals.plos.org/plosone/s/submission-guidelines#loc-laboratory-protocols

We look forward to receiving your revised manuscript.

Kind regards,

Peter M ten Klooster, Ph.D.

Academic Editor

PLOS ONE

Journal Requirements:

1. Thank you for including your competing interests statement; "PT, MC, WT have declared no competing interests exist. KR has received funding from ASICS, outside the submitted work. Prof Dalbeth reports research grant funding from Amgen and AstraZeneca, and speaker fees from Pfizer Inc, Horizon, Janssen Pharmaceuticals, and AbbVie, as well as consulting fees from Horizon, Hengrui, and Kowa, outside the submitted work."

2. Please include captions for your Supporting Information files at the end of your manuscript, and update any in-text citations to match accordingly. Please see our Supporting Information guidelines for more information: http://journals.plos.org/plosone/s/supporting-information

Reviewers' comments:

Reviewer's Responses to Questions

**Comments to the Author**

1. Is the manuscript technically sound, and do the data support the conclusions?

Reviewer #1: Yes

Reviewer #2: Yes

2. Has the statistical analysis been performed appropriately and rigorously? 

Reviewer #1: Yes

Reviewer #2: Yes

3. Have the authors made all data underlying the findings in their manuscript fully available?

Reviewer #1: Yes

Reviewer #2: Yes

4. Is the manuscript presented in an intelligible fashion and written in standard English?

Reviewer #1: Yes

Reviewer #2: Yes

5. Review Comments to the Author

Reviewer #1: This Delphi study about features of retail shoes for women with rheumatoid arthritis is very important and strongly needed. The study is well designed consisted experts worldwide. Couple of minor points to revise:

-Background is very short. Can you consider broadening it to take into account footwear adherence and related factors.

-Dot is missing before the aim of the study at the end of Background.

-Through the text you separate the UK from Europe, to my knowledge UK is part of Europe. Moreover, in text and Table 1 location is based on coutry or continent - can you unify these? Maybe report countries where the experts came?

-Maybe one recommendation could also be for footwear manufacturer? And inclusion of the to future research projects.

Thank you, very good study and well written.

Reviewer #2: Thank you for the opportunity to review this manuscript. It is of interest to researchers and health professionals who provide foot care and advice for people who have RA.

My comments are minor and primarily relating to some of the grammar only.

Abstract - research is performed with participants rather than on participants. Please consider modifying to:

An international Delphi exercise using online survey software was conducted with 39 participants from health care backgrounds.

Also, there are multiple sentences that are missing full stops. Please reread and correct.

Introduction:

Consider amending for plural consistency: whilst offering the woman a choice in style

To: whilst offering women choices in style

Methods:

Amending: A sample of health professionals was

To: A sample of health professionals were

Expertise - based on your inclusion criteria - I would suggest that the expertise was footwear prescription rather than just footwear?

At times in the methods section there are tense issues and would urge the authors to use past tense throughout

Results:

female with mean (SD) of 23 (10) mean years - delete duplicate mean

I note that the methods describe presenting the 30th and 70th percentile, IPA and then IQR? Please present one of the other or describe why these have been used in each section.

6. PLOS authors have the option to publish the peer review history of their article (what does this mean?). If published, this will include your full peer review and any attached files.

Reviewer #1: No

Reviewer #2: Yes: Cylie M Williams

---

## [Decision Letter · Decision Letter 1]

10 Dec 2019

Important features of retail shoes for women with rheumatoid arthritis: A Delphi consensus survey

PONE-D-19-30180R1

Dear Dr. Tehan,

We are pleased to inform you that your manuscript has been judged scientifically suitable for publication and will be formally accepted for publication once it complies with all outstanding technical requirements.

With kind regards,

Peter M ten Klooster, Ph.D.

Academic Editor

PLOS ONE

Additional Editor Comments (optional):

Reviewers' comments:

Reviewer's Responses to Questions

**Comments to the Author**

1. If the authors have adequately addressed your comments raised in a previous round of review and you feel that this manuscript is now acceptable for publication, you may indicate that here to bypass the “Comments to the Author” section, enter your conflict of interest statement in the “Confidential to Editor” section, and submit your "Accept" recommendation.

Reviewer #1: All comments have been addressed

Reviewer #2: All comments have been addressed

2. Is the manuscript technically sound, and do the data support the conclusions?

Reviewer #1: Yes

Reviewer #2: Yes

3. Has the statistical analysis been performed appropriately and rigorously? 

Reviewer #1: Yes

Reviewer #2: Yes

4. Have the authors made all data underlying the findings in their manuscript fully available?

Reviewer #1: Yes

Reviewer #2: Yes

5. Is the manuscript presented in an intelligible fashion and written in standard English?

Reviewer #1: Yes

Reviewer #2: Yes

6. Review Comments to the Author

Reviewer #1: (No Response)

Reviewer #2: Thank you for providing these changes. I only have one minor grammar change that I recommend based on the additions and that is to add the word ‘wearing or recommendations’ to the sentence following Spence as you can’t adhere to footwear but I wear to wearing or recommendations.

Lack of adherence with therapeutic footwear is multi-factorial, often compounded by a lack of acceptance of their underlying condition, and a disparity in expectations of the footwear

7. PLOS authors have the option to publish the peer review history of their article (what does this mean?). If published, this will include your full peer review and any attached files.

Reviewer #1: Yes: Minna Stolt

Reviewer #2: Yes: Cylie M Williams

---

## [Editor Report · Acceptance letter]

17 Dec 2019

PONE-D-19-30180R1 

Important features of retail shoes for women with rheumatoid arthritis: A Delphi consensus survey 

Dear Dr. Tehan:

I am pleased to inform you that your manuscript has been deemed suitable for publication in PLOS ONE. Congratulations! Your manuscript is now with our production department. 

With kind regards,

on behalf of

Dr. Peter M ten Klooster 

Academic Editor

PLOS ONE